# Improvement in Corrosion Resistance and Interfacial Contact Resistance Properties of 316L Stainless Steel by Coating with Cr, Ti Co-Doped Amorphous Carbon Films in the Environment of the PEMFCs

**DOI:** 10.3390/molecules28062821

**Published:** 2023-03-21

**Authors:** Baosen Mi, Quan Wang, Yuhao Xu, Ziwei Qin, Zhuo Chen, Hongbin Wang

**Affiliations:** 1School of Materials Science and Engineering, Shanghai University, Shanghai 200444, China; 2Panxing Technology Zhejiang Co., Ltd., Jinhua 321000, China; 3Institute of Materials Science, Shanghai DianJi University, Shanghai 201306, China; 4Shanghai Engineering Research Centre for Metal Parts Green Remanufacture, Shanghai 200444, China

**Keywords:** amorphous carbon films, Cr, Ti co-doped, interfacial contact resistance, corrosion resistance, bipolar plate

## Abstract

In order to obtain films with high corrosion resistance and excellent interfacial contact resistance (ICR) on 316L stainless steel used for bipolar plates in proton-exchange membrane fuel cells (PEMFCs), Cr, Ti co-doped amorphous carbon films were prepared on 316L stainless steel. The preparation method for the coating was magnetron sputtering. The doping amount of the Ti element was controlled by a Cr target and a Ti target current. The change in the structure and properties of the coating after the change from Cr single-element doping to Cr and Ti co-doping was studied. The change rule of the structure and properties of the coating from Cr single-element doping to Cr and Ti co-doping was studied. An increase in the Ti content led to a decreased grain boundary, a flatter surface, and a higher sp^2^-hybridized carbon content. TiC and CrC nanocrystals were formed in the amorphous carbon structure together. The amorphous carbon films doped with Cr and Ti simultaneously achieved a low ICR and high corrosion resistance compared with single-Cr-doped amorphous carbon. The enhanced corrosion resistance was attributed to the decreasing grain boundary, the formation of the TiC crystal structure, and the smaller grain size. The best performance was obtained at a Ti target current of 2A. Compared with bare 316L stainless steel, the corrosion resistance of Cr, Ti co-doped amorphous carbon (I_corr_ = 5.7 × 10^−8^ A/cm^2^, Ti-2 sample) was greatly improved. Because Ti doping increased the content of sp^2^-hybridized carbon in the coating, the contact resistance of the coating decreased. Moreover, the interfacial contact resistance was 3.1 mΩ·cm^2^ in the Ti-2 sample, much lower than that of bare 316L stainless steel. After the potentiostatic polarization test, the coating still had excellent conductivity.

## 1. Introduction

Proton-exchange membrane fuel cells (PEMFCs) are obtaining much attention due to the urgent need for environmental protection. Bipolar plates form the backbone of the PEMFC stack and constitute 30% of the cost of PEMFCs [1,2]. Bipolar plates have several functions, such as collecting the current generated during the operation of the battery pack, separating oxygen and hydrogen, limiting gas leakage, and acting as a transmission channel for the reaction gas and product water [3]. Bipolar plates should have a low interfacial contact resistance (ICR) and high corrosion resistance during the operation of PEMFCs. The US Department of Energy (DOE) target for corrosion currents density is ≤1 μA/cm^2^, and the target for ICR is ≤10 mΩ·cm^2^. 

Graphite is often used to manufacture bipolar plates. The properties of graphite bipolar plates allow them to meet most of the bipolar plates’ demands, for instance, good corrosion resistance and electrical conductivity. However, graphite bipolar plates are expensive, as they have high manufacturing costs. Furthermore, graphite has poor mechanical properties [4]. Thus, researchers have been focusing on developing appropriate substitute materials for graphite to address these challenges. 

Metal bipolar plates, such as steel, titanium, copper, aluminum, and several alloy systems, are more commonly used for PEMFCs due to their low cost, high mechanical properties, and excellent conductivities [5,6,7,8,9]. Stainless steel is a desirable choice for PEMFC bipolar plates due to its reasonably low cost, high mechanical strength, and wide availability [10]. However, stainless steel cannot satisfy the requirements of PEMFC applications. It was revealed that stainless steel is vulnerable to corrosion in the acidic environment found in PEMFCs, and the steel becomes covered with oxide layers. However, the oxide layers drastically improved the stainless steel’s ICR [11].

Significant attention has been devoted to developing suitable coatings to improve corrosion resistance and ICR by mitigating these problems. Some surface engineering techniques exist for applying coatings, including chemical vapor deposition, magnetron sputtering, and electroplating [12,13,14]. Among the choices, magnetron sputtering is a desirable method for manufacturing various films on stainless steel. The performance of materials, such as transition metal nitrides, nanocomposite and amorphous carbon films, to improve the corrosion resistance and ICR vales of the bipolar plates of fuel cells was investigated [15,16,17,18]. Amorphous carbon films, which have excellent chemical stability and electrical conductivity, have been attracting extensive attention as potential films for stainless steel bipolar plates [19]. However, in PEMFC applications, amorphous carbon film-coated bipolar plates always become damaged during stack operation due to the unavoidable coating corrosion deterioration and increased ICR. Many researchers have tried to retard the film damage by doping elements into pure amorphous carbon films [20,21,22,23]. The multiple elements of the amorphous carbon films may have a severe impact on the performance of the sputtering coating. For example, Wang et al. [20] studied the corrosion behavior of 304 stainless steel bipolar plates protected with Cr-doped carbon coatings applied by magnetron sputtering, testing the films in the PEMFC environment. Chromium doping increased the sp^2^-hybridized carbon content, and the Cr doping also increased the density of the amorphous carbon. The authors reported that the Cr-doped amorphous carbon exhibited excellent corrosion resistance and ICR values compared with the 304 stainless steel. Yan et al. [24] reported results obtained by applying a silver nanowire-doped amorphous carbon film to stainless steel and aluminum bipolar plates. The authors found that the anti-corrosion properties improved significantly after Ag was doped into the amorphous carbon film. Jia et al. [25] studied Cr and Cu co-doped carbon films. They reported that the amount of Cu and Cr doped in the amorphous carbon was critical for the structure of the amorphous carbon film. Proper Cr and Cu doping improved the sp^2^-hybridized carbon in the amorphous carbon, which is good for contact resistance. The Cu- and Cr-deposited ions increased gradually, enhancing the densification of the film. Specifically, the sp^2^/sp^3^ hybridized carbon ratio and the compactness of the film affected the performance of the amorphous carbon film stainless steel. The density and sp^2^/sp^3^ hybridized carbon ratio of the amorphous carbon coating can be further modified by elemental doping, which can improve the application of the coating in fuel cells, enhancing the performance of the fuel cells’ bipolar plate coatings. 

Although some studies have analyzed the structure and performance of binary-doped amorphous carbon coatings, few studies have been applied to the bipolar plates of proton-exchange membrane fuel cells. At the same time, the research on the effect of Cr and Ti co-doping on the performance of amorphous carbon coatings applied to fuel cell bipolar plates has received little attention. In this work, a series of Cr, Ti co-doped amorphous carbon films were deposited on 316L stainless steel substrates using magnetron sputtering. The amorphous carbon film’s structure was adjusted by changing the Ti target current. Mainly, the structural transformation law of the coating from single-element Cr doping to binary Cr, Ti doping was studied. The effects of the Ti doping amount on the properties of Cr, Ti binary-doped amorphous carbon coatings were analyzed. In the application environment of the coating on the bipolar plate of the proton-exchange membrane fuel cell, the corrosion resistance effect and the change rule of the contact resistance during the structural transformation were analyzed. The relationship between the structure and properties of amorphous carbon doped by single-element Cr, and Cr- Ti binary elements was revealed in this study. This work may provide new ideas about designing high-performance amorphous carbon films for stainless steel bipolar plates in PEMFCs.

## 2. Results and Discussion

### 2.1. Morphology

Scanning electron microscopy (SEM) was used to examine the Cr, Ti-doped amorphous carbon film’s surface morphology on the stainless steel. Figure 1a shows the micrograph of the Cr-doped amorphous carbon film. The surface shows a clear grain boundary and rugged surface. Figure 1b,d shows the micrograph of the Cr, Ti-doped amorphous carbon. The grain boundary and grain size of the amorphous carbon films decreased with the Ti incorporation, which was beneficial for corrosive improvement. When the Ti target increased to 3 A, the surface became rough, which may have been caused by increased sputtering energy. Moreover, it was evident that the surface morphology of all samples became tighter, and no apparent defects were found. However, the growth of the columnar structure roughness was promoted when the Ti target current was increased to 3A, which had an adverse effect. Figure 1e presents the SEM images of the cross-section of the Ti-2 film. The three layers comprising different components are clearly visible. No microcracks were observed at the layer interfaces. 

XPS analyzed the bonding structure of all the amorphous carbon films. XPS peak fit software was used to analyze the curve; the Shirley background subtraction method with the Gaussian–Lorentz function method was used to fit the peaks to obtain the chemical bond data. Figure 2 shows the XPS spectra of the deposited Ti, Cr co-doped amorphous carbon films with different Ti target currents. Figure 2a shows that the peak of C 1s was deconvoluted into three or four main components at ~282.3 eV, ~282.8 eV, ~284.2 eV, ~285.1 eV, and ~288.2 eV, which indicated the existence of Ti-C, Cr-C, sp^2^, sp^3^, and C=O peaks, respectively [26,27,28]. There were four σ bonds in the case of the sp^3^-hybridized carbon, which were beneficial to the anti-corrosion properties and the mechanical properties of the amorphous carbon film. In the sp^2^-hybridized carbon, an σ bond made with three electrons and a π bond, which are good for the electronic properties [19]. In the Cr 2p spectrum shown in Figure 2b, two peaks corresponding to the Cr-C bond were identified, the peaks were observed at approximately ~574.1 eV and ~574.8 eV (according to the NIST database), whereas another corresponding to the Cr-O bond was observed at ~576.7 eV, which represents the Cr-O bond. The existence of the Cr-O peak indicates some chromium oxide (CrO) in the air environment [20]. Figure 2c exhibits the Ti 2p curve, which deconvoluted two small peaks. The Ti-C bond, which, according to the peaks at ~454.7 eV and ~455.5 eV, separately suggested that when Ti was doped into the amorphous carbon film, TiC formed in the film, and there were no simple atomic titanium or Ti clusters in the film [29,30]. The peak at 455.5 eV here was related to the carbon-rich carbon phase Ti-C [26,31]. Figure 2a reveals that the Ti-C phase’s peak area increased, which indicates that the contents of the Ti-C phase of the prepared thin films improved gradually with the increase in the Ti target current. The ratio of sp^2^/sp^3^ was calculated from the C1s in XPS and is shown in Figure 2d. It was found that the ratio of sp^2^/sp^3^ in the coating increased with the increase in Ti content. The Ti content in the amorphous carbon was calculated by XPS, and the results are shown in Figure 2d. It was found that with an increase in the Ti target current, the content of Ti in the amorphous carbon increased slowly (0–2 A) at first and then rapidly (2–3 A). When the sputtering current was small, the ions with lower energies bombard the surface, resulting in fewer atoms being splashed out. Increasing the target current increased the energy of the sputtering ions bombarding the surface, increasing the number of sputtering Ti atoms. At the same time, it has been reported that large numbers of Cr and C atoms in the magnetron sputtering cavity also react with and are deposited on the Ti surface. Although the amount of this deposition is relatively small, it exists at a low target current. At this time, the target surface experiences two processes: sputtering and deposition. When the current of the Ti target is low, more deposition reactions occur on the target surface, and fewer Ti atoms are sputtered out. By contrast, when the current increases to a certain extent, the sputtering of Ti dominates, and more Ti atoms are sputtered out, resulting in a significant increase in the content of Ti in the coating [32]. This is the main reason for the large change in the Ti content in the coating. 

To further study the changes in the amorphous carbon structure, the Raman spectra of these four coatings were measured. Figure 3a shows the four-coating Raman spectra, and the ID/IG and G peaks are shown in Figure 3b, which were fitted using the Gaussian function. The Raman spectrum of the amorphous carbon can be fitted to obtain two peaks: the D peak (located at 1320–1360 cm^−1^) and the G peak (located at 1520–1600 cm^−1^). They are attributed to the breathing mode of the sp^2^ carbon rings (D peak) and the stretching vibration of the C-C sp^2^-hybridized carbon (G peak) [33,34].

The positions of the ID/IG and G peaks are closely related to the state of the sp^2^-hybridized carbon. The ID/IG ratio corresponds to the amount of sp^2^-hybridized carbon. The G peak position is related to the topological disorder of the carbon film caused by the size and shape of the sp^2^ cluster [35]. As shown in Figure 3b, the ID/IG and G peaks shifted to a higher value with an increase in the Ti content in the coating. The results indicate that the amount of the sp^2^-hybridization carbon in the film also increased with the increase in the content of Ti. The shift of the G peak position to a higher value indicates that the sp^2^-hybridized carbon’s size and degree of ordering increased [28]. This is because Ti atoms can easily bond with carbon, which reduces the suspended bond in the amorphous carbon structure. Ti doping into the amorphous carbon catalyzes the sp^3^-to-sp^2^ transformation, which increases the sp^2^-hybridized carbon content in the amorphous carbon film [31,36]. 

The results of the XRD analysis for different samples are depicted in Figure 4. The XRD patterns of all the layers exhibited the peaks of the Ti, CrN, and 316L stainless steel substrate. This is because the penetration depth of the X-ray was larger than the layer’s thickness, causing diffraction from the CrN layer, Ti basement layer, and steel substrate. The analysis of the XRD results confirmed the presence of the CrC phase on the amorphous carbon film. Additionally, two short and small TiC peaks were observed in the XRD diffraction pattern after Ti doping into the Cr-doped amorphous carbon. The TiC peaks confirmed the presence of a crystal structure. 

Detailed structures of the films were analyzed from the TEM results with corresponding selected area electron diffraction (SAED) patterns. The Ti-0 sample TEM and the Inverse Fast Fourier Transform (IFFT) images obtained from the corresponding selected area are shown in Figure 5e,f. It was confirmed that the film was composed of the Cr-C nanocrystal, which is marked by a white oval. The size was about 10 nm in width. The nanocrystal was surrounded by amorphous carbon. The HRTEM images and SAED pictures of the Ti, Cr co-doped amorphous carbon are displayed in Figure 5a–d. It can be seen that the nanocrystal structure is still surrounded by amorphous carbon. The incorporation of Ti does not change the embedding form of the nanocrystals. In the SAED diagram, in contrast to Ti-0, TiC nanocrystals appeared in the coating after the introduction of the Ti atoms. Furthermore, the appearance and change in the TiC nanostructure affected the performance of the coating.

Combined with the XPS, Raman, TEM, and XRD results, the TiC phase appeared in the coating after titanium was doped into the amorphous carbon, and the TiC phase existed in the form of a crystal structure. When Ti was doped into the amorphous carbon, the content of the sp^2^-hybridized carbon in the coating increased.

### 2.2. The Electrochemical Corrosion Behavior

Figure 6 demonstrates the potentiodynamic polarization curves. The different corrosion current densities and corrosion potential values of the amorphous carbon and 316L stainless steel are shown in Figure 6. Table 1 shows the relevant values obtained from the potentiodynamic polarization curves. As shown in Figure 6 and Table 1, uncoated stainless steel exhibited the highest corrosion current density, 6.98 × 10^−5^ A/cm^2^. A lower corrosion current density represents better corrosion resistance. When the Cr and Cr, Ti-doped amorphous carbon films prepared on the steel surface acted as good corrosion resistance barriers, the corrosion current density decreased sharply. The conclusion suggests that the element-doped amorphous carbon coating can considerably improve the corrosion resistance of 316L stainless steel. The Ti-0 sample’s corrosion current density was 7.6 × 10^−8^ A/cm^2^. When Ti atoms were doped into the coating to form amorphous carbon co-doped with Cr and Ti, the corrosion current density was further reduced. The Ti-1 sample’s current corrosion current was 6.6 × 10^−8^ A/cm^2^, and the Ti-2 sample had the lowest corrosion current density, 5.7 × 10^−8^ A/cm^2^. However, when the Ti content in the coating increased further, the corrosion current density of the coating also increased to 1.57 × 10^−7^ A/cm^2^ (Ti-3 sample). The increase in the corrosion current density indicates that when the content of Ti in the amorphous carbon reached a certain level, the corrosion resistance of the coating was adversely affected. When titanium is doped into amorphous carbon, the difference in titanium content results in a change in its surface morphology. 

The SEM images, which are shown in Figure 1, exhibited that the grain size of the film became smaller when Ti was doped into the amorphous carbon. The doping of titanium reduced the gap between grains, thus reducing the channel of the corrosion solution entering the matrix, which increased the corrosion resistance of the amorphous carbon coating. A further increase in the Ti content would change the density of the amorphous carbon coating, thus reducing the corrosion resistance. In addition, the appearance of the TiC crystal phase also increases the corrosion resistance of the coating after Ti doping because the TiC phase in amorphous carbon often exists as a corrosion-resistant phase [26]. 

The corrosion current density results indicated that the corrosion resistance of the amorphous carbon coating increased when the CrC nanocrystalline was transformed into two kinds of nanocrystalline structures, CrC and TiC. A decrease in the corrosion resistance was caused by the excessive doping of Ti atoms, which might also be related to an increase in the sp^2^-hybridized carbon in the coating. The Raman results indicated that an increase in the Ti content led to an increase in the sp^2^-hybridized carbon content in the coating. Ti-3 had the highest content of sp^2^ and a higher range of sp^2^, which had a negative effect on the corrosion resistance. For the Ti-1 and Ti-2 samples, although the content of the sp^2^-hybridized carbon increased, the impact of the TiC phase reduced the growth of the sp^2^-hybridized carbon content.

Figure 7 presents polarization curves of the Cr, Ti co-doped amorphous carbon. The potentiostatic polarization curves of all the samples except Ti-0 at 0.6 V (vs. SCE) showed a rapid downward trend. After 2000 s, the corrosion current density was gradually stabilized. The constant corrosion current density was related to the stable, protective film on the surface [37]. In the final stable corrosion current density, the amorphous carbon coating co-doped with Cr and Ti (Ti-1, Ti-2, and Ti-3 samples) still exhibited a lower value than the single Cr-doped amorphous carbon coating (Ti-0). Regarding the potentiodynamic curve, the Ti-1 sample demonstrated low and unstable anti-corrosion resistance properties due to the density and structure of the coating. The results implied that in the potentiostatic polarization test, the amorphous carbon coating co-doped with Cr and Ti exhibited a better corrosion resistance than the Cr-doped amorphous carbon coating. The Ti-2 sample had the lowest corrosion density, approximately 9.062 × 10^−9^ A/cm^2^ (+0.6 V (vs. SCE)) and 5.353 × 10^−8^ A/cm^2^ (+1.1 V (vs. SHE)). Combined with the test results of the potentiodynamic polarization curve and potentiostatic polarization curve, Ti-2 exhibited a higher corrosion resistance than the other samples.

### 2.3. Interfacial Contact Resistance (ICR)

Figure 8a shows the curve of the contact resistance versus the pressure of different samples. The contact resistance decreased with an increase in pressure. This is related to the contact area between the sample and the carbon paper. With an increase in pressure, the contact resistance between the sample and the carbon paper gradually increased, resulting in a decrease in contact resistance. The ICR of the 316L stainless steel, Cr-doped amorphous carbon-coated samples, and the Cr, Ti co-doped amorphous carbon films were measured under a 1.4 MPa compression force, as shown in Figure 8b. The bare 316L stainless steel had an ICR of 132.8 mΩ·cm^2^. Stainless steel forms a passive film in the atmosphere, and the passive film dramatically increases the ICR values between the bipolar plate and the carbon paper. When the amorphous carbon coating with excellent conductivity replaces the passive film, the ICR value of the coating is greatly reduced. After the Ti atoms were doped into amorphous carbon, the Cr, Ti co-doped amorphous carbon was formed, the sp^2^ hybridization bonds were increased, and some TiC crystal structures were formed in the amorphous carbon film. As a conductive phase in the amorphous carbon, the increase in the sp^2^-hybridized carbon can improve the conductivity of the coating. At the same time, as a conductive crystal with low resistivity, TiC can also improve the conductivity of the coating. The ICR value decreased from 3.9 mΩ·cm^2^ (Ti-0) to 3.5 mΩ·cm^2^ (Ti-1). Typically, the Ti-2 sample exhibited a slightly lower ICR in all the coated samples; the best ICR value obtained in this work was 3.1 mΩ·cm^2^. Doping with a high Ti content did not improve the ICR performance of the films, which was 9.9 mΩ·cm^2^ in the Ti-3 sample; this could be related to the oxidation of metal atoms into oxides in the air when there are too many doped metal elements. In conclusion, these ICR values provide valuable insights into the engineering demands to further lower the ICR values of 316L stainless steel bipolar plates by designing Cr, Ti co-doped amorphous carbon films. As shown in Figure 8, the ICR value of the coated samples after +1.1 V (vs.SHE) potentiostatic polarization did not increase significantly. By contrast, the ICR value of the Ti-2 samples increased from 3.1 mΩ·cm^2^ to 4 mΩ·cm^2^.

The chemical characterization of all coatings after +1.1 V (vs. SHE) electrochemical corrosion was performed using XPS to observe the change in the chemical state of the coating before and after corrosion (Figure 9). The XPS curve results revealed that the C, CrC, and TiC bonds suffered slightly oxidation. Some C-O bond and Ti-O bond peaks appeared on the XPS curves, proving that some C-O and Ti-O formed on the amorphous carbon surface. On the other hand, the Cr-O peak w clearly increased on the Cr 2p XPS curve, especially for the Ti-0 and Ti-2 samples, which is similar to our previous work [38]. Finally, the ICR values increased after the corrosion due to the formation of oxide products.

## 3. Experimental Section

### 3.1. Sample Preparation

The substrate material was 316L stainless steel. The coating substrates were 50 mm × 50 mm squares with a thickness of 0.1 mm. They were ultrasonically cleaned in deionized water, anhydrous ethanol, and acetone, each for 15 min, prior to loading into the sputtering chamber. The coating samples were synthesized by magnetron sputtering. Four magnetron targets were installed in the vacuum magnetron sputtering chamber. One magnetron target was set up with carbon targets, which were used to create the carbon composition. Two magnetron targets were fitted with chromium targets to produce the Cr doping element in the amorphous carbon and as the element source of the CrN interfacial layer. The one remaining target was loaded on a titanium target to produce the basement Ti layer. The Ti target was also used for doping Ti atoms into the carbon layer. The sputtering gases, nitrogen and argon with a purity of 99.99%, were admitted into the sputtering chamber. The deposition process for the carbon film followed the four steps below. The first step was ion cleaning. A high bias voltage of up to 500 V was applied to the steel substrates to etch the surface. The argon flow rate was 20 sccm, which was maintained until the end of coating preparation. During coating preparation, the pressure inside the cavity was 1.77 × 10^−1^ Pa. This step aimed to remove the contamination of the 316L stainless steel surface. Second, the Ti basement layer was built. A pure Ti layer was first built on the substrate. Third, the CrN interfacial layer was built. A CrN interfacial layer was deposited next by gradually reducing the deposition rate of Ti while progressively increasing the deposition rate of Cr. In this step, nitrogen, which was the source of the nitrogen atoms, was introduced into the chamber. The nitrogen flow rate was 20 sccm in this step. Fourth, the Cr, Ti co-doped amorphous carbon layer was built. The Cr, Ti co-doped top layer was deposited. By controlling the target current on the Ti target, the Ti doping content in the Cr-doped amorphous carbon was tuned. In this step, nitrogen flow was closed, the current of the C target was 6 A, and the currents of two Cr targets were 1 A. The Ti target currents were set to 0 A, 1 A, 2 A, and 3 A. The samples were named Ti-0, Ti-1, Ti-2, and Ti-3 according to the Ti target current. With the increase in the Ti current, the content of Ti element in the coating increased gradually, and there were no Ti atoms doped in the Ti-0 samples. 

### 3.2. Samples Characterization

The surface and section morphologies of all films were characterized by scanning electron microscopy (Sigma 300, Zeiss, Germany). The chemical bonds of the Cr, Ti co-doped amorphous carbon top layers were analyzed by X-ray photoelectron spectroscopy (XPS, Escalab 250Xi, Thermo Scientific Escalab, America). The excitation source was an Al kα ray (hv = 1486.6 eV) with an energy step of 0.05eV. The vacuum degree of the vacuum chamber tested by XPS was 8 × 10^−10^ Pa. X-ray diffraction patterns were obtained using a D/MAX2200V X-ray diffractometer instrument (Rigaku Corporation, Japan) to obtain the crystallographic features of the layers. XRD analysis was conducted using Cu kα radiation over 2θ from 30 to 80° with a scan rate of 4°/min. Raman spectroscopy (inVia Qontor, Renishaw, Britain)was employed to analyze the amorphous carbon structure on the surface of films with a 532 nm laser wavelength. Transmission electron microscopy (TEM, JEM-2010, JEOL, Japan) was applied to characterize the microstructure of the films. The top amorphous carbon layer, which was used for the TEM test, was cut by FIB cutting.

### 3.3. Electrochemical Corrosion

The Cr, Ti co-doped amorphous carbon film and the uncoated 316L stainless steel were loaded into a three-electrode electrochemical work station (Reference 600, Gamry, America) with a coated sample exposed area of 1 cm^2^. A 0.5 mol/L H_2_SO_4_ and 2 ppm HF solution were used as the corrosive solution, and the solution was set at 70 °C to simulate the PEMFC operation environment. The reference and counter electrodes of the Ag/AgCl and platinum plates, respectively, were used. The potentiodynamic polarization curves of all the plates were measured from −0.3 V (vs. SCE) to 1.2 V (vs. SCE) of the potential range with a scan rate of 1 mV/s. Moreover, the potentiostatic polarization test was conducted at +0.6 V (vs. SCE) and +1.1 V (vs. SHE) potential for 7200 s. 

### 3.4. Interfacial Contact Resistance (ICR)

The ICR between the bipolar plate and carbon paper was been tested according to a procedure mentioned in other works, which is the optimized method of Wang [24,39]. Two pieces of carbon paper (Toray TGP-H-060) were used.

## 4. Conclusions

In this study, Cr and Ti co-doped amorphous carbon was prepared on 316L stainless steel by magnetron sputtering. The ICR and corrosion resistance were tested in the PEMFC environment. The content of the Ti atom as a variable was controlled by the current of the titanium target. The content of the Ti atoms increased gradually with the increase in the Ti target current, and it decreased the boundary of the columnar crystal. The incorporation of Ti made the nanocrystalline CrC in the coating change into CrC and TiC nanocrystalline without changing the amorphous carbon structure of the coating. The Cr, Ti co-doped amorphous carbon films exhibited a lower corrosion current density. A low corrosion current density value (5.7 × 10^−8^ A/cm^2^ at Ti-2 sample) of the coated 316L stainless steel bipolar plates used in PEMFCs was achieved by an amorphous carbon film doped with Cr and Ti. After Ti doping into amorphous carbon, the formation of TiC and the decrease in the grain boundary contributed to the low corrosion current density. Both Cr-doped and Cr, Ti co-doped amorphous carbon films showed a lower ICR than bare 316L stainless steel. The Cr, Ti co-doped amorphous carbon film with a Ti target current of 2A showed the lowest ICR value of 3.1 mΩ·cm^2^. The lowest ICR was mainly attributed to the appropriate TiC crystal structure and the enhanced sp^2^-hybridized carbon in the coating. Doping a certain amount of Ti can enhance the compactness of Cr-doped amorphous carbon films and ICR stability. The ICR of the Ti-2 sample, measured after the corrosion test, was 4 mΩ·cm^2^, showing excellent conductivity.

## Figures and Tables

**Figure 1 molecules-28-02821-f001:**
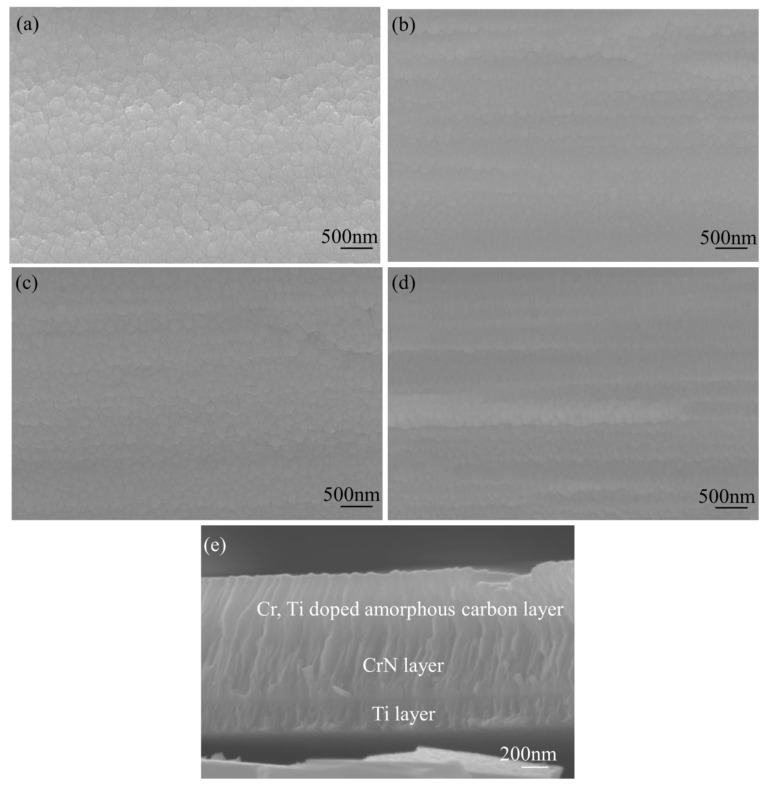
SEM images showing the changing of the Cr, Ti-doped amorphous carbon coating for different Ti concentrations; (**a**) Ti-0, (**b**) Ti-1, (**c**) Ti-2, (**d**) Ti-3, (**e**) cross-section of the Ti-2 film.

**Figure 2 molecules-28-02821-f002:**
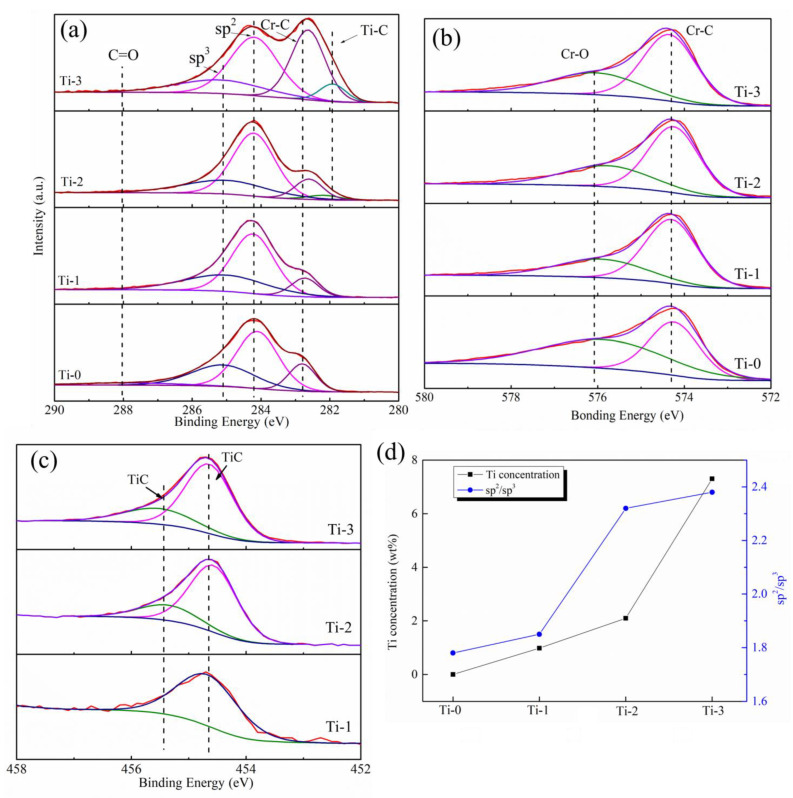
XPS patterns of the as-deposited composite amorphous carbon films; (**a**) C1s, (**b**) Cr 2p, (**c**) Ti 2p, (**d**) Ti content of different Ti target currents.

**Figure 3 molecules-28-02821-f003:**
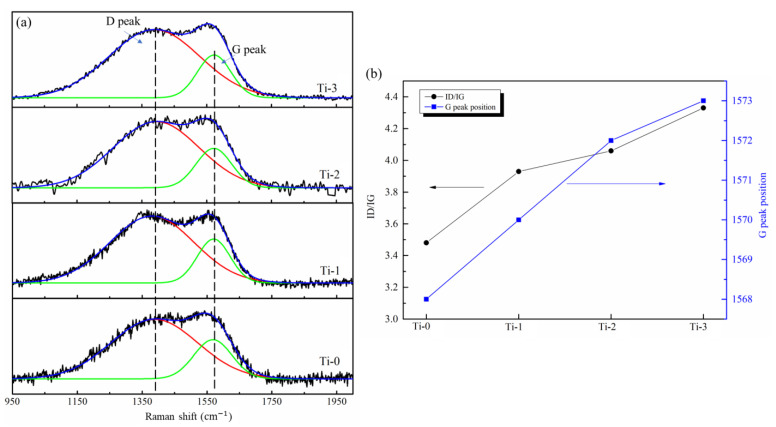
Raman spectra of amorphous carbon films: (**a**) Raman spectra (**b**) ID/IG and G peak position.

**Figure 4 molecules-28-02821-f004:**
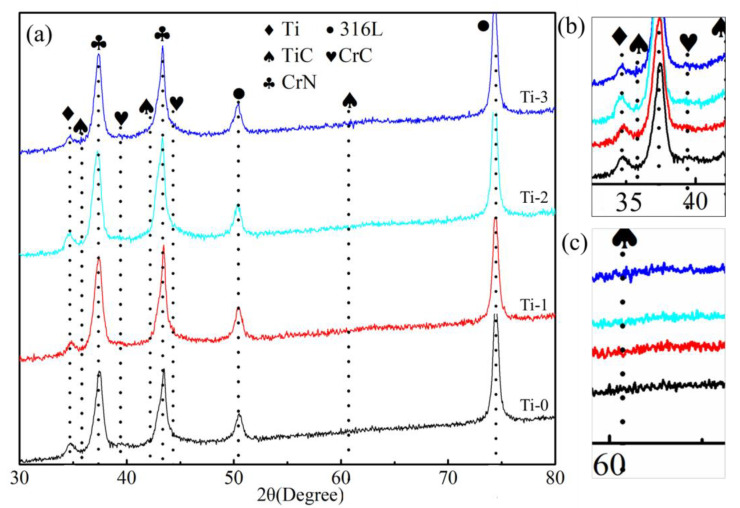
XRD analysis results for all coated specimens: (**a**) XRD spectrum, (**b**,**c**) partial enlarged view of TiC.

**Figure 5 molecules-28-02821-f005:**
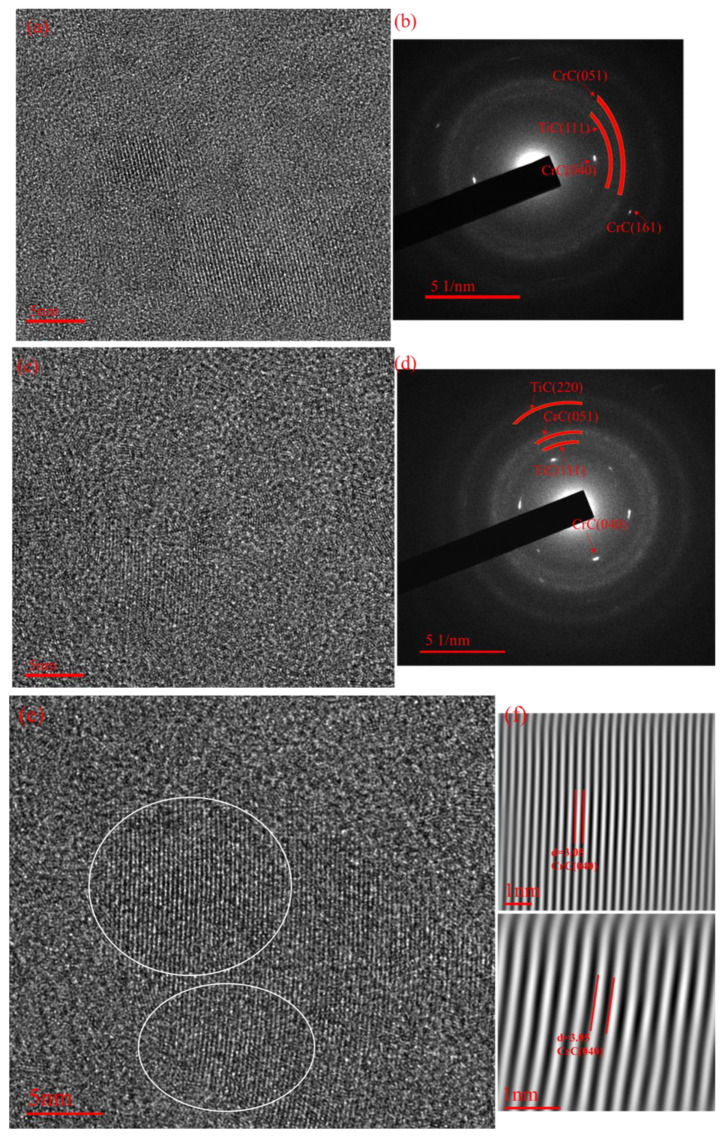
TEM images and inset SAED patterns of the Ti, Cr co-doped amorphous carbon film: (**a**,**b**) Ti-1, (**c**,**d**) Ti-2, (**e**,**f**) Ti-0.

**Figure 6 molecules-28-02821-f006:**
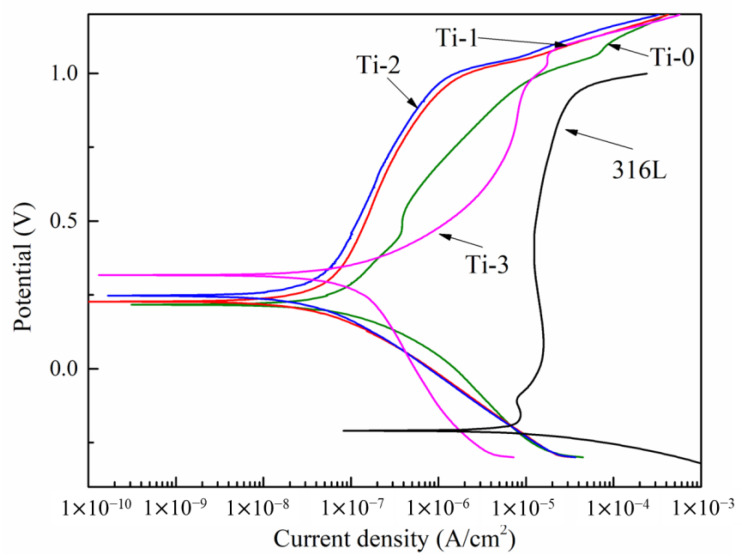
Potentiodynamic polarization curves of sample in the environment of PEMFCs.

**Figure 7 molecules-28-02821-f007:**
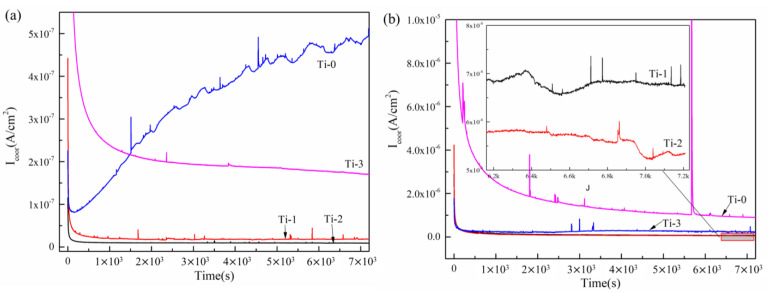
Potentiostatic polarization curves of Cr, Ti co-doped amorphous carbon film: (**a**) +0.6 V (vs. SCE), (**b**) +1.1 V (vs. SHE).

**Figure 8 molecules-28-02821-f008:**
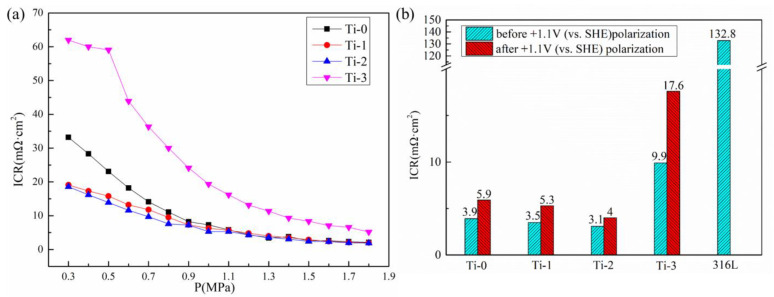
The ICR values of bare 316L stainless steel and coated samples. (**a**) Contact resistance under different pressures, (**b**) ICR values at 1.4 MPa.

**Figure 9 molecules-28-02821-f009:**
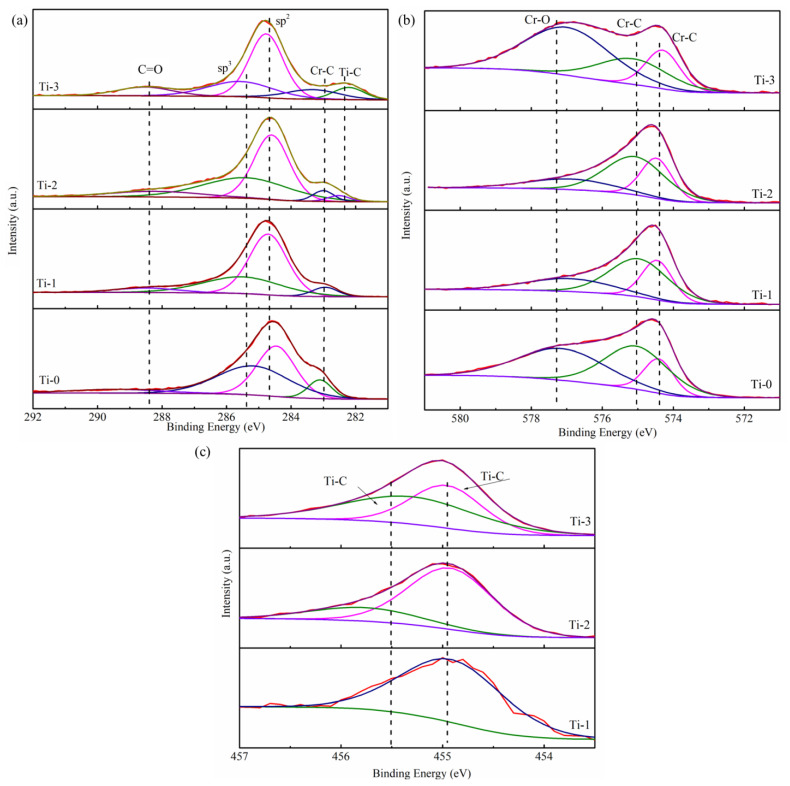
XPS patterns of the amorphous carbon films after +1.1 V (vs. SHE) corrosion, (**a**) C1s, (**b**) Cr 2p, (**c**) Ti 2p.

**Table 1 molecules-28-02821-t001:** Electrochemical parameters of all the samples calculated by potentiodynamic polarization curves.

Samples	Steel	Ti-0	Ti-1	Ti-2	Ti-3
I_corr_ (μA/cm^2^)	69.8	0.076	0.066	0.057	0.152
E_corr_ (mV vs. SCE)	−216	203	227	247	317

## Data Availability

The data generated or analyzed during the study are included in the article.

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
