# Peer review of "Improvement in Corrosion Resistance and Interfacial Contact Resistance Properties of 316L Stainless Steel by Coating with Cr, Ti Co-Doped Amorphous Carbon Films in the Environment of the PEMFCs"

_molecules, 2023, doi:10.3390/molecules28062821_

Round 1
Reviewer 1 Report
Overview:
In molecules-2254275, authors report on the structure, corrosion resistance and interfacial contact resistance of a-C:Cr:Ti material formed by multi-target magnetron sputtering. Some characteristics relevant for the films’ application as coatings of bipolar plates of PEMFCs are assessed. Although authors present scientifically sound results, the discussion of the structural investigations and several other issues should be significantly revised prior to the publication.
Major comments:
1) In their study, authors assess the electrochemical corrosion behavior and ICR of the films. However, they haven’t yet reported the application of the films in the real PEMFCs. There are also several characteristics required to be measured to verify if the reported films are practically applicable, such as in-plane electrical conductivity and flexural strength [Planes et al. Energy Procedia 20:311-323. 10.1016/j.egypro.2012.03.031]. Therefore, I suggest that the claims such as “…for application in bipolar plates of PEMFCs” (lines 4-5), “…used for bipolar plates in PEMFCs.” (line 351) should be revised, and explicit statement on current stage of the studies should be made: what has been done, are the obtained results promising or not in comparison to other coatings, what should be done next to assess the films applicability in the suggested area?
2) Lines 74-75: “Chromium doping increases the sp2 bond content and density of amorphous carbon.”; lines 168-170: “The microstructure images show that after Ti was doped, the grain boundary of the columnar structure decreased, which indicates that the coating density has been improved.”: generally, the variation of the a-C films’ density is associated with sp2/sp3 ratio (see Fig. 11 in Ferrari et al. Phys. Rev. B 62, 11089 DOI 10.1103/PhysRevB.62.11089]) and hydrogen content (Table 1 in [Kaplan et al., Appl. Phys. Lett. 47, 750 (1985); https://doi.org/10.1063/1.96027]). There are ways to measure films’ density, for example, by deriving it from the position of the C1s π+σ plasmon loss satellite. However, authors don’t use them, and – in my opinion – unreasonably claim that lower concentration of grain boundaries resulted in higher density despite sp2-hybridized carbon fraction rise. Therefore, I suggest that all density-related fragments should be revised by introducing more appropriate term to discuss the evolution of the samples (for example, ordering/disordering/amorphization).
3) Lines 118-119 indicate that “The sputtering gases, nitrogen and argon with a purity of 99.99% were admitted into the sputtering chamber.”; this resulted in the formation of CrN sublayer, but authors claim that the investigated upper layer isn’t nitridized despite being deposited in nitrogen-containing atmosphere. Is it really the case? Presenting the elemental composition assessed via XPS would benefit the discussion on the nitrogen content. Additionally, taking the nitrogen into account may fix some issues with XPS spectra fitting (see below).
4) In Fig. 2, XPS reveals various widths of fitting components, which is not typically the case. First, it is not enough information to discuss this data, as the type of baseline and details of the fitting are not provided (software, line types, parameters of the fitting). This data should be added in the revised manuscript. Second, can this issue be resolved by the introduction of the new fitting components related to the nitrogen content, if N is present in the samples?
5) Is Raman data indicating sp2-carbon content increase with the increased Ti flux confirmed by the variation of the intensity ratio of sp2/sp3 XPS components? Do any of the data make it possible to determine if the samples are diamond-like (ta-C) or graphite-like (a-C)? Please keep in mind that sp2-rich subsurface layer is typical for PVD-deposited carbon-based films, which can affect the results obtained by the XPS (see [Streletskiy at al. Applied Physics A volume 128, Article number: 929 (2022) DOI 10.1007/s00339-022-06062-2]).
Minor comments:
6) Current titles of the samples are inconsistent, for example, sample is named “Ti-2A” in line 177, while it is “Ti-2” in line 27. Such naming is also confusing to the readers, as Ti is apparently not a predominant element in the samples. In my opinion, the coatings have structure commonly referred to as a-C:Cr/a-C:Cr:Ti. Information about the samples’ elemental composition which can be found out from the XPS data will make it possible to provide informative titles like “a-C:Cr10%:Ti10%”.
7) Line 82 and below, “improved the sp2 bond in amorphous, which”: the “sp2 bond” is common term, but it is a slang one: the sp2/sp3 hybridization is a characteristic of carbon atom, not a bond. The fragments operating “sp2/sp3 bonds” terminology should be revised.
8) Lines 118-119, “The sputtering gases, nitrogen and argon with a purity of 99.99% were admitted into the sputtering chamber.”: what was the ratio of the gases flux? What were the residual and work pressure?
9) Line 138, “A monochromatic Al Kα line was used to acquire the spectra with an energy of 1486.76 eV”: did you mean the diffraction patterns rather than spectra? Or this line is referred to XPS? What were the operating parameters of XPS spectrometer?
10) Lines 140-141: “Raman spectroscopy was employed to analyze the graphite structure on the surface of films”: why not diamond structure? Did you mean that Raman spectroscopy is predominantly sensitive to the sp2-hybridized carbon ([Streletskiy at al. Applied Physics A volume 128, Article number: 929 (2022) DOI 10.1007/s00339-022-06062-2])? Fragment needs revision.
11) Lines 166-167: “Moreover, it is evident that all the samples’ surface has dense, and compact structure and no apparent defects were found.”: what do you mean by “compact structure”?
12) Line 184: 288.2 eV is a position of C=O, O–C–O, and N–C=N bonding (see [Zavidovskii et al., Journal of Experimental and Theoretical Physics volume 134, pages 682–692 (2022) DOI 10.1134/S1063776122050144] and refs. 34-35 within) rather than C-O. The 286-287 eV position are typical for C-O (see [Chen et al., Fullerenes, Nanotubes and Carbon Nanostructures Volume 28, 2020, Issue 12 DOI 10.1080/1536383X.2020.1794851]).
13) Lines 186-189: “In one sp3 bond, there is a σ bond built with four valence electrons, which are benefited to the anti-corrosion property and mechanical property of amorphous carbon film. In the sp2 bond, a σ bond made with three electrons, and a π bond with a fourth free electron, is good for the electronic properties[19]”: please indicate that there are 4 σ bonds in case of sp3-hybridized carbon, while 3 σ bonds and one π bond “form” in case of sp2 carbon.
14) Fig. 2a: the color of the sp2 fitting component varies for the Ti-3 spectra, is this done on purpose?
15) Line 194: could you please explain why there are 2 Ti-C peaks in Ti2p fitting (~454.7 eV and ~455.1 eV) in more detail?
16) Line 196: by “simple titanium metal”, did you mean “atomic titanium” or “Ti clusters”?
17) Lines 201-202: “the target surface will undergo two steps: sputtering and deposition”: what do you mean by “target surface deposition”? Line 203: what did you mean by “deposition reactions”? Did you mean that sputtering is followed by the condensation of the particles, or the contribution of the re-sputtering process? Overall fragment needs a revision.
18) Lines 213-214: “The Raman spectrum of amorphous carbon can be fitted to obtain two peaks: G peak (around 1350 cm-1) and D peak (approximately 1556 cm-1)”: D-peak is located at 1320–1360 cm−1, and G-peak is located 1520–1600 cm−1 (see for example [Streletskiy et al., Magnetochemistry 2022, 8(12), 171; https://doi.org/10.3390/magnetochemistry8120171]).
19) Lines 214-215: “They are related to the breathing mode of the sp2 bond (D peak) and the elongation of the sp2 bond in the graphite crystal structure in amorphous carbon.[33, 34]”: how can a bond have a breathing mode? You apparently meant the breathing mode of graphitic hexagon. By elongation, did you mean the stretching vibrations of the C=C bonds? Appropriate discussion is provided on page 9 in [Streletskiy at al. Applied Physics A volume 128, Article number: 929 (2022) DOI 10.1007/s00339-022-06062-2].
20) Line “222”, “more graphite crystal”: first, are there graphitic crystallites in the amorphous matrix? What technique shows that? Second, do you mean that the crystallites become more graphite-like (i.e., less disordered/more pristine), or that larger amount of the crystallites is formed in the matrix?
21) Line 226, what do you mean by “suspended bond”? If the unsaturated bonds are being saturated after Ti incorporation, that should lead to the sp2→sp3 rearrangement.
22) In Fig. 4, could you make an inset demonstrating the emergence of TiC-related peaks? Currently they are not visible in the diffraction patterns, but smoothing and background subtraction may help.
23) Line 243: define FFT. Additionally, FFT of the image should provide a SEAD-like pattern: are you sure that the results of FFT are presented in (f)?
24) According to the Fig.1, the thickness of the Ti/CrN/a-C:Cr:Ti heterostructure is close to 1 mkm, which is generally too thick for the TEM studies (Fig. 5). Please describe carefully, how were the samples prepared for TEM, and what kind of samples were investigated via TEM (a-C:Cr:Ti or Ti/CrN/a-C:Cr:Ti)?
25) In Fig. 6, the arrow overlaps with “Ti-0” caption.
26) Lines 292-293: “From the Raman results, the growth of Ti content leads to increased sp2 bond content in the coating and increased graphitization degree in the coating.”, could you clarify what is the difference between the sp2 content increase and graphitization of the coating?
27) In Fig. 7, “x” should be replaced with “×” in Y scale, “1k” and “Time (s)” should be replaced with “103” and “Time (s)” or “1” and “Time (103×s). Additionally, the X and Y scale of the inset of Fig. 7b interferes with the lines of the main subfigure.
28) As for the Fig. 7(a), could you explain/suggest why does the potentiostatic polarization curve of titan-free sample differs from the one of the titan-containing samples?
Reviewer 2 Report
(1) Different multilayers of Ti/(Ti, Cr) N/CrN , Cr/C, and Cr /Ti have been used to coat 316L stainless steel, what's the novelty of this research?
(a) Tassin, C., Laroudie, F., Pons, M., & Lelait, L. (1996). Improvement of the wear resistance of 316L stainless steel by laser surface alloying. Surface and Coatings Technology, 80(1-2), 207-210.
(b) Li, P., Dong, H., Xia, Y., Hao, X., Wang, S., Pan, L., & Zhou, J. (2018). Inhomogeneous interface structure and mechanical properties of rotary friction welded TC4 titanium alloy/316L stainless steel joints. Journal of Manufacturing Processes, 33, 54-63.
(c) Wang, S., Hou, M., Zhao, Q., Jiang, Y., Wang, Z., Li, H., ... & Shao, Z. (2017). Ti/(Ti, Cr) N/CrN multilayer coated 316L stainless steel by arc ion plating as bipolar plates for proton exchange membrane fuel cells. Journal of energy chemistry, 26(1), 168-174.
(2) Many pictures should be modified carefully. They're randomly drawn.
(3) "When Ti is doped into amorphous carbon, the content of the sp2 bond in the coating increase"? Why and how? Clarify in the manuscript.
It needs some revisions.
Reviewer 3 Report
In this manuscript, authors decrease the corrosion resistance and ICR of 316L stainless steel by a novel coating technique, which is of significant importance in improving the PEMFC performance. I would like to recommend acceptance of this manuscript after several minor revisions, as below:
1. Page 1, in the introduction section, in fact, the functions of BP are not limited to collect the current, …, gas supply and liquid removal are also the main functions of BP, authors can refer to Chem. Rev. 2023, 123, 989−1039; J. Renewable Sustainable Energy 2021, 13, 022701;
2. It is recommended that authors compare the ICR at more compression forces. Only the data at 1.4 MPa is provided in Fig. 8;
3. It will be better that authors present the polarization curves of the PEMFC using the coated BP.
Round 2
Reviewer 1 Report
Authors have addressed the majority of my comments in accurate manner, and the manuscript was improved substantially. I have a few more suggestions for the revised text.
1) Line 4 (Title), by “environment PEMFCs”, did you mean “PEMFCs environment”/“environment of the PEMFCs”?
2) Line 125: “1.33×10-3 Toor”: first, “Torr”; second, SI unit are preferrable.
2) From Fig.1 and Authors’ reply, I understand what they mean by “increased the density of amorphous carbon” (line 75) and “enhancing the densification of the film” (line 84). However, as I have outlined in Major comment 2, density of the film decreases with the increase of sp2-crabon content, which was observed by authors with Ti content rise. As the density was not measured throughout the research, I suggest revising the “densification” term: the films’ structure became tighter/less loose/more spatially uniform, its clustering improved; however, the films did not become “denser”.
3) In Fig. 2(a), replace “C-O” with “C=O” in accordance with the revised text.
4) Lines 188-189, please revise the sentence “The Shirley background correction was used on the fit progress which select a proper range.”, as it is currently unclear.
5) Lines 230-231, “G peak (located at 1320–1360 cm−1) and D peak (located at 1520–1600 cm−1).”: once again, D peak is located at 1320–1360 cm−1, and G peak is located at 1520–1600 cm−1. Please revise the fragment.
Reviewer 2 Report
(1) "there is no simple atomic titanium or Ti clusters in the fi The bipolar plate has many functions..."? Rephrase it.
(2) Moderate English changes are required.
